# Age prediction from coronary angiography using a deep neural network: Age as a potential label to extract prognosis-related imaging features

**Shinnosuke Sawano**[1], **Satoshi Kodera**[1]*, **Masataka Sato**[1], **Susumu Katsushika**[1], **Issei Sukeda**[1], **Hirotoshi Takeuchi**[1], **Hiroki Shinohara**[1], **Atsushi Kobayashi**[1], **Hiroshi Takiguchi**[1], **Kazutoshi Hirose**[1], **Tatsuya Kamon**[1], **Akihito Saito**[1], **Hiroyuki Kiriyama**[1], **Mizuki Miura**[1], **Shun Minatsuki**[1], **Hironobu Kikuchi**[1], **Yasutomi Higashikuni**[1], **Norifumi Takeda**[1], **Katsuhito Fujiu**[1,2], **Jiro Ando**[1], **Hiroshi Akazawa**[1], **Hiroyuki Morita**[1], **Issei Komuro**[1]

**1** Department of Cardiovascular Medicine, The University of Tokyo Hospital, Tokyo, Japan, **2** Department of Advanced Cardiology, The University of Tokyo, Tokyo, Japan

\* koderasatoshi@gmail.com

**Data Availability Statement:** The data cannot be made public due to the protection of the patients' personal information. Data are available from the cardiovascular medicine in the University of Tokyo

## Abstract

Coronary angiography (CAG) is still considered the reference standard for coronary artery assessment, especially in the treatment of acute coronary syndrome (ACS). Although aging causes changes in coronary arteries, the age-related imaging features on CAG and their prognostic relevance have not been fully characterized. We hypothesized that a deep neural network (DNN) model could be trained to estimate vascular age only using CAG and that this age prediction from CAG could show significant associations with clinical outcomes of ACS. A DNN was trained to estimate vascular age using ten separate frames from each of 5,923 CAG videos from 572 patients. It was then tested on 1,437 CAG videos from 144 patients. Subsequently, 298 ACS patients who underwent percutaneous coronary intervention (PCI) were analysed to assess whether predicted age by DNN was associated with clinical outcomes. Age predicted as a continuous variable showed mean absolute error of 4 years with R squared of 0.72 ($r = 0.856$). Among the ACS patients stratified by predicted age from CAG images before PCI, major adverse cardiovascular events (MACE) were more frequently observed in the older vascular age group than in the younger vascular age group (p = 0.017). Furthermore, after controlling for actual age, gender, peak creatine kinase, and history of heart failure, the older vascular age group independently suffered from more MACE (hazard ratio 2.14, 95% CI 1.07 to 4.29, p = 0.032). The vascular age estimated based on CAG imaging by DNN showed high predictive value. The age predicted from CAG images by DNN could have significant associations with clinical outcomes in patients with ACS.

Hospital (contact via kayo.cho@gmail.com - the person in charge of data management at the Department of Cardiovascular Medicine, University of Tokyo Hospital) for researchers who meet the criteria for access to confidential data.

**Funding:** The author(s) received no specific funding for this work.

**Competing interests:** The authors have declared that no competing interests exist.

## Introduction

Coronary artery angiography (CAG) is still considered the reference standard for definitive diagnosis of coronary artery disease [1], especially for the treatment of acute coronary syndrome (ACS) [2, 3], even though non-invasive testing has become more widespread [4]. However, CAG is an invasive test with the risk of complications such as bleeding and stroke [5], and when it is performed, it is therefore desirable to obtain as much useful information as possible for patient evaluation. To address this issue, we focused on the estimation of vascular age using CAG imaging in this study. Vascular age is a concept in relation to the hypothesis that the conversion of chronological age to age derived from vascular imaging features will lead to more accurate assessment of an individual's cardiovascular risk. Although coronary artery calcium (CAC) scoring by computed tomography (CT) [6] and carotid intima-media thickness (CIMT) [7] assessment by B-mode ultrasonography can be used to define vascular age [8], it is not clearly known whether CAG contains useful age-related imaging features. Recently, deep neural networks (DNNs) have been utilized to analyse various types of images [1, 9], including data interpretation that is difficult for humans, such as predicting age and gender from electrocardiograms [10]. The purpose of this study was to develop a deep learning neural network to estimate vascular age based on coronary angiographic imaging and to examine the clinical usefulness of this age prediction.

## Material and methods

### Study design and coronary artery angiography acquisition

The study design was a single-centre retrospective analysis. Consecutive patients aged ≥18 years who underwent standby diagnostic CAG for any reason between January 2010 and December 2015 at The University of Tokyo Hospital were reviewed. Patients undergoing PCI or CABG after index CAG were included in the study. Patients who underwent PCI with diagnostic CAG (i.e., ad-hoc PCI) were excluded. **CAGs that a** highly trained cardiologist **judged as showing insufficient contrast effect were excluded** after image processing, and, if a wire or balloon used for PCI or measurement of fractional flow reserve was captured on the images, these images were manually excluded, as were contrast images of coronary artery bypass. There was no other information in the image like the ECG signal or date of birth that could have also been used by the neural network to estimate age. Only those CAGs evaluated as "without coronary artery stenosis greater than 75%" were included. All CAG procedures in the included patients were performed as standard procedures in The University of Tokyo Hospital. This study was conducted in accordance with the revised Declaration of Helsinki and was approved by our institutional and local ethics committees (reference number 2650-(13)). Informed consent was obtained in the form of an opt-out selection on the web-site.

### Cohort for neural network training, validation, and testing

A total of 7,360 videos from 937 CAG procedures performed in 716 patients were randomly divided into a training dataset (4,771 CAG videos from 457 patients [63.8%]), validation dataset (1,152 CAG videos from 115 patients [16.1%]), and test dataset (1,437 CAG videos from 144 patients [20.1%]). No patients were included in more than one of the three datasets. The training dataset was used to train the neural network and the hyper parameters were then tuned using the validation dataset. The prediction accuracy of the final neural network was tested using the test dataset.

## Coronary artery angiography and chronological age datasets

CAG videos of different time length were acquired as Digital Imaging and Communication in Medicine (Dicom) files recorded at 15 frames per second. The image size was $512 \times 512$ pixels and each pixel contained density information from 0 to 255. Therefore, the structure of a CAG video was the 3D matrix ($T_{nk}$, $Y_{ni}$, $X_{nj}$), where $_{nk}$ represents the number of frames in each CAG video, $_{ni}$ is the pixel position from 0 to 512, and $_{nj}$ is the pixel position from 0 to 512. Since some CAG videos contained frames with high information and others with low information, the frames with high information content were extracted using edge filters [11, 12]. A total of ten frames were used from each video, with the frames with the largest edge being selected, and each final video was represented by a matrix of (10, 512, 512).

The patients' chronological ages, defined as the number of years since birth, were obtained from the catheterization report.

## Development of the neural network for vascular age estimation

First, we defined vascular age as the age estimated based on a patient's CAG imaging by the neural network. A two-dimensional convolutional neural network (2D-CNN) to predict age from CAG was implemented using transfer learning and fine-tuning techniques in Python [13] with 4 sets of Nvidia Tesla A100 80 GB graphics processing unit (NVIDIA Corporation, Santa Clara, USA). Briefly, the methods utilized a pre-trained neural network to reduce the training time and the amount of data required for training. A neural network can learn much faster and with substantially fewer training examples if transfer learning and fine-tuning are employed, rather than training from scratch [14, 15]. We adopted the EfficientNet [16], a commonly used 2D-CNN architecture, as the pre-trained neural network. The pre-trained weights on ImageNet for EfficientNet were downloaded from https://github.com/Cadene/pretrained-models.pytorch [17].

A single video represented by a (10, 512, 512) matrix was treated as ten separate frames, each of which was given a chronological age label and reshaped to (10, 600, 600) for input into the neural network. The training dataset was used to train the neural network to predict age as a continuous variable by minimizing the mean squared error (MSE) between predictions and ground truth age labels. A classification neural network to detect age ≥65 was also created using a similar neural network with binary cross-entropy loss [18]. These procedures used an Adam optimizer [19] with a batch size of 16 for 100 epochs. The learning rate was set to an initial value of 0.00001, 0.000001, 0.0000001, or 0.0000005 and then gradually reduced, with the initial learning rate with the lowest MSE or binary cross-entropy loss at the time of inference being used. The learning rate was reduced by a factor of two if the validation loss plateaued after three epochs. If the loss did not decrease for five consecutive epochs, the neural network training was stopped, even if 100 epochs had not been completed, and the neural network weights at the lowest validation loss were saved.

## Performance evaluation

The trained neural network was applied to the CAG images in the test dataset and the predicted ages were calculated as continuous variables. These test dataset predictions were used to evaluate the predictive performance of the neural network on a per-CAG procedure basis. The per-frame assessment depended on the results of a single-frame, the per video assessment was the average of the ten per-frame assessments, and the per-CAG assessment was the average of the per-video assessments [20]. The correlation coefficient R, R squared, and mean average error were used to evaluate the neural network. The outputs of the neural network were also evaluated as multi-group to determine the accuracy of the predicted age within the age groups of 18 to 50, 50 to 70, and over 70 years. For classification neural network to detect age ≥65, the

accuracy, sensitivity, and specificity with a cut-off value of 0.5, and the area under the receiver operating characteristics curve, were calculated. Additional subgroup analyses were performed for target coronary artery (right or left) and gender (male or female). The gradient-weighted class activation mapping (Grad-CAM) method was used to visualize the regions affecting the interpretations of the developed neural network [21].

## Associations between estimated vascular age and clinical outcome in patients with ACS

Associations between predicted age and clinical outcomes in patients with ACS were examined. This analysis included 298 ACS patients who underwent PCI at our institution between 2010 and 2015 and whose video acquisitions were available. ACS was defined according to the universal definition [3]. The exclusion criteria were: (1) the second or more than second PCI performed during the study period, (2) patients with a history of coronary artery bypass grafting, (3) patients without follow-up information. Finally, 298 ACS patients were used to evaluate the associations between predicted age and clinical outcomes in patients with ACS. All individual CAG images were evaluated using a network pre-trained to estimate vascular age. The predicted age was obtained from each pre-PCI image as a continuous variable, and ACS patients were divided into two groups: a younger vascular age group (predicted age <65) and an older vascular age group (predicted age ≧65) [22]. The major adverse cardiovascular events (MACE) were compared between a younger vascular age group and an older vascular age group. MACE were defined as cardiac death, ACS, non-fatal cerebral infarction, and admission for heart failure. The index date was the date when the PCI was performed.

Hypertension was defined as a systolic blood pressure >140 mmHg, diastolic blood pressure >90 mmHg, or medical treatment for hypertension [23]. Diabetes mellitus was defined as haemoglobin A1c >6.5% or treatment for diabetes mellitus [24]. Dyslipidaemia was defined as total cholesterol >220 mg/dl, low-density lipoprotein cholesterol >140 mg/dl, or treatment for hyperlipidaemia. Shock was defined as systolic blood pressure <90 mmHg, use of vasopressors to maintain blood pressure, or attempted cardiopulmonary resuscitation [24]. Cerebral infarction was defined as an acute episode of focal or global neurological dysfunction caused by brain, spinal cord, or retinal vascular injury resulting from haemorrhage or infarction [25].

## Statistical analysis

Data are expressed as mean ± standard deviation or number (percentage). Categorical variables were compared using the chi squared test (or Fisher's exact test for small samples). Normally distributed continuous variables were compared using Student's $t$ test and abnormally distributed continuous variables were compared using the Mann–Whitney U test. Event free survival curves were constructed using the Kaplan–Meier method, and statistical differences between curves were assessed using the log-lank test. P values < 0.05 were considered statistically significant. A multivariate Cox regression analysis was performed to investigate associations between in-hospital complications and MACE after controlling for known clinical confounders. Hazard ratios (HRs) and 95% confidence intervals (CI) were calculated. All statistical analyses were performed using R (R Foundation for Statistical Computing, Vienna, Austria).

## Results

### Patient selection

A total of 7,360 CAG videos from 937 CAG procedures performed on 716 patients between January 2010 and December 2015 were included. In total, 106 patients underwent multiple

**Table 1. Patient and vessel characteristics at the time of coronary artery angiography.**

| Variables | Training dataset | Validation dataset | Test dataset | P value |
|---|---|---|---|---|
| **Cases of CAG (n = 937)** | n = 603 | n = 152 | n = 182 | |
| Age, (years) | 57.8 ± 17.3 | 55.5 ± 17.9 | 57.3 ± 17.2 | 0.363 |
| Age, (groups) | | | | |
| < 40 | 112 (18.6%) | 28 (18.4%) | 33 (18.1%) | 0.541 |
| 40–49 | 80 (13.3%) | 30 (19.7%) | 21 (11.5%) | |
| 50–59 | 83 (13.8%) | 24 (15.8%) | 31 (17.0%) | |
| 60–69 | 143 (23.7%) | 27 (17.8%) | 46 (25.3%) | |
| 70–79 | 146 (24.2%) | 35 (23.0%) | 39 (21.4%) | |
| 80+ | 39 (6.5%) | 8 (5.3%) | 12 (6.6%) | |
| Sex | | | | |
| female, n (%) | 233 (38.6%) | 49 (32.2%) | 53 (29.1%) | 0.039 |
| male, n (%) | 370 (61.4%) | 103 (67.8%) | 129 (70.9%) | |
| Body height, (cm) | 162.2 ± 12.0 | 161.9 ± 15.5 | 163.9 ± 9.13 | 0.239 |
| Body weight, (kg) | 60.2 ± 14.4 | 60.2 ± 13.9 | 61.5 ± 13.5 | 0.531 |
| **Vessel (n = 7360 videos)** | n = 4,771 videos | n = 1,152 videos | n = 1,437 videos | |
| RCA | 1628 (34.1%) | 387 (33.6%) | 496 (34.5%) | 0.886 |
| LCA | 3143 (65.9%) | 765 (66.4%) | 941 (65.5%) | |

Data are expressed as mean ± standard deviation or number (percentage).

Pearson's chi-square test was used for categorical variables, Student's *t*-test was used for normally distributed continuous variables, and the Mann–Whitney U test was used for non-normally distributed continuous variables. RCA, right coronary artery; LCA, left coronary artery

CAG procedures. The characteristics of the patients included in this study are shown in Table 1. The mean age of the study population was 57.3 ± 17.4 years (minimum 18 years, maximum 90 years). There were 602 CAG procedures with 4,738 videos from men and 335 CAG procedures with 2,622 videos from women. We enrolled 4,849 left coronary artery (LCA) videos and 2,511 right coronary artery (RCA) videos in this study. The training, validation, and test datasets included 4,771 CAG videos from 457 patients (63.8%), 1,152 CAG videos from 115 patients (16.1%), and 1,437 CAG videos from 144 patients (20.1%), respectively (Fig 1, Table 1).

## Performance in the age prediction

As the output of the neural network was a continuous variable, the statistic of absolute error was calculated together with the overall correlation and the explained variance (R squared). For the test dataset, the mean absolute error was 4 years and R squared was 0.72 ($r = 0.856$). A scatter plot of chronological age versus predicted age is presented in Fig 2A. For the multi-group classification into age groups of 18 to 50, 50 to 70, and 70 years and above, the overall accuracy was 68% (Fig 2B). For detection of age ≥65, the AUC was 0.839 with a sensitivity of 74%, specificity of 76%, and accuracy of 75% (S1 Fig). Subgroup analysis according to target vessel showed R squared of 0.69 ($r = 0.830$) in the RCA group and 0.73 ($r = 0.846$) in the LCA group (Fig 3A and 3B). Gender analysis showed R squared of 0.68 ($r = 0.826$) in the male group and 0.83 ($r = 0.910$) in the female group (Fig 3C and 3D).

## Visualization of neural network decision making

Grad-CAM analysis demonstrated that the neural network focused on the entire coronary artery limbus to predict age from CAG (Fig 4).

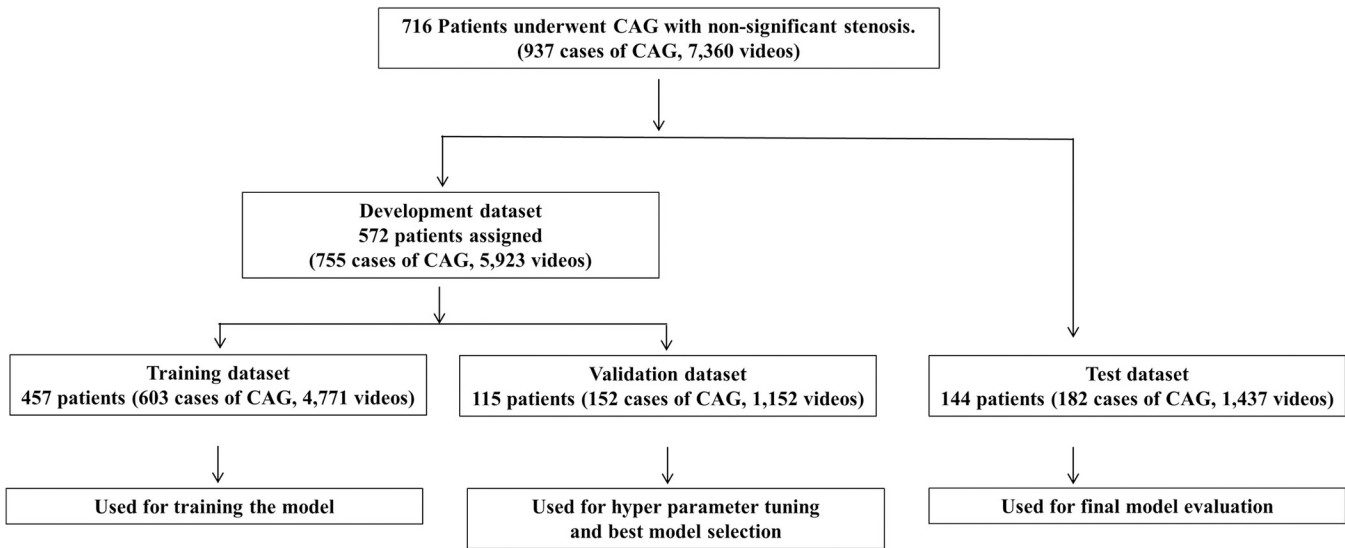

**Fig 1. Patient selection.** From a total of 7,360 coronary artery angiography (CAG) videos in 716 patients, 572 patients (5923 videos) were randomly allocated to the development dataset. This dataset was further split into 457 patients (4,771 videos) for training and 115 patients (1,152 videos) for validation. The remaining 144 patients (1437 videos) were allocated to the test dataset. The model was trained solely on CAG videos from the training dataset. Hyper parameter tuning and selection of the best model within 10 epochs was performed using the validation dataset. The test dataset was used solely for testing the performance of the final model. There were no overlaps in patients between the three datasets.

## Associations between predicted age and clinical outcomes in patients with ACS

The clinical characteristics of the younger vascular age group and older vascular age group are shown in Table 2 for the 298 ACS patients used to determine associations between predicted age obtained from pre-PCI CAG images and clinical outcomes. The mean absolute error between predicted age and chronological age was 3 years with an R squared of 0.38 ($r = 0.615$). ST-elevated myocardial infarction, male sex, and peak-CK were higher in the younger vascular age group than in the older vascular age group, and the percentage of chronological age $\geqq 65$ years was 37.7% in the younger vascular age group and 75.5% in the older vascular age group (Table 2). The clinical outcomes of the two groups are shown in Table 3. MACE were more frequently observed in the older vascular age group (41 of 184, 22.3%) than in the younger

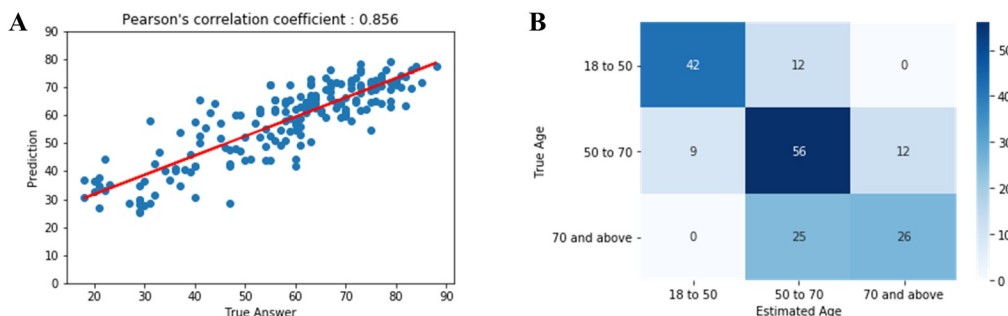

**Fig 2. Predicted age based on CAG imaging versus chronological age. A** Estimated vascular age versus reported chronological age (in years; red, regression line). R squared for the model was 0.72 with a Pearson correlation of $r = 0.856$. **B** Patients were classified by age (in years) into groups of 18 to 50, 50 to 70, and over 70 years, and the number of patients of a given actual age (y axis) classified into each estimated vascular age (x axis) was shown to confirm accuracy. The overall accuracy was 68%.

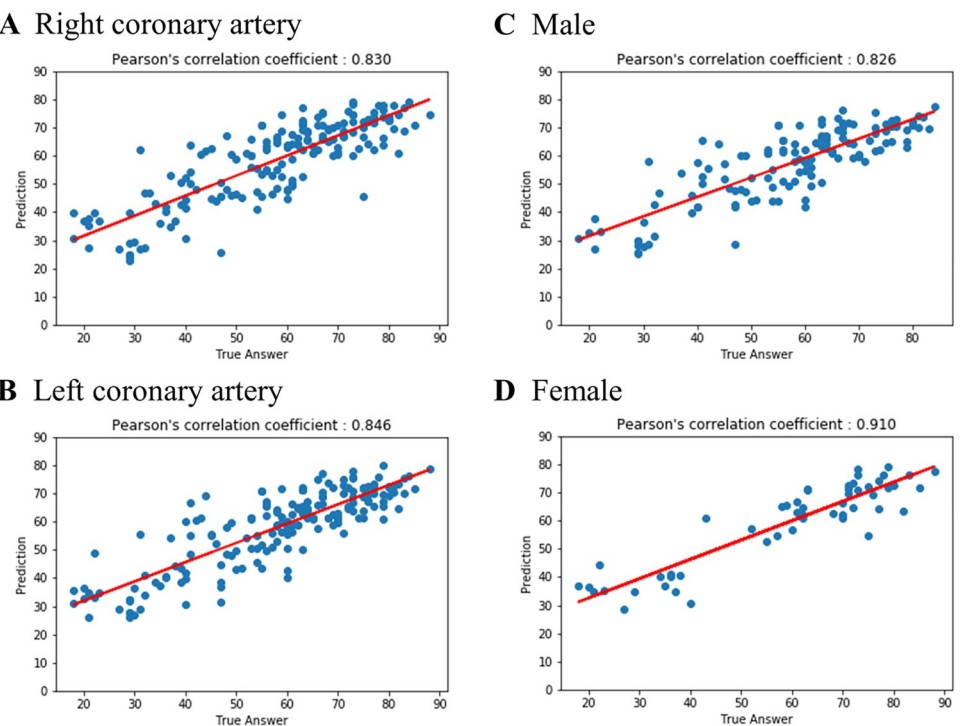

**Fig 3. Predicted age based on CAG imaging versus chronological age in the subgroup analysis.** Shown is the estimated vascular age versus the reported chronological age in the subgroup analysis (in years; red, regression line). A, right coronary artery; B, left coronary artery; C, male; D, female.

vascular age group (14 of 114, 12.3%)(p = 0.031). Fig 5 shows Kaplan-Meier curves for MACE in the two groups. The median follow-up duration was 1893 days. The MACE were more frequently observed in the older vascular age group than in younger vascular age group (P = 0.017).

Results of the multivariate Cox regression analysis are presented in Table 4. The older vascular age group showed a significant association with MACE (hazard ratio 2.14, 95% CI 1.07

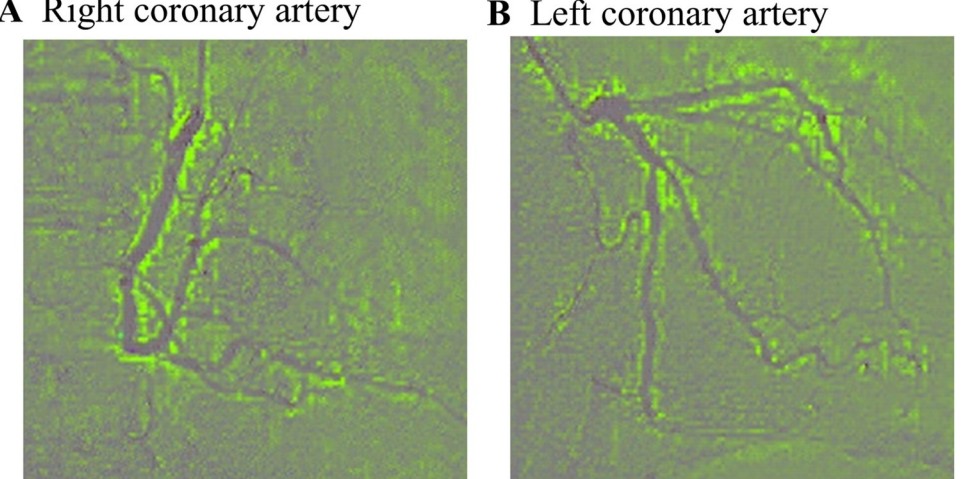

**Fig 4. Representative Grad-CAM images (Efficientnet-B7, using the test dataset).** Green areas represent the areas upon which the model focused. A, right coronary artery; B, left coronary artery.

**Table 2. Comparison of clinical characteristics between the younger vascular age group and older vascular age group.**

| Variables | All (n = 298) | Younger vascular age group (n = 114) | Older vascular age group (n = 184) | P value |
|---|---|---|---|---|
| Chronological Age, (years) | 67 ± 12 | 60 ± 13 | 71 ± 10 | <0.001 |
| Chronological Age, (≧65 years) | 182 (61.1%) | 43 (37.7%) | 139 (75.5%) | <0.001 |
| male sex | 228 (76.5%) | 102 (89.5%) | 126 (68.5%) | <0.001 |
| Body Mass Index, (kg/m$^2$) | 24 ± 3.8 | 24.5 ± 4.0 | 23.9 ± 3.6 | 0.134 |
| ST elevated myocardial infarction | 145 (48.7%) | 76 (66.7%) | 69 (37.5%) | <0.001 |
| Hypertension | 207 (69.5%) | 65 (57.0%) | 142 (77.2%) | <0.001 |
| Diabetes mellitus | 98 (32.9%) | 34 (29.8%) | 64 (34.8%) | 0.376 |
| Dyslipidaemia | 183 (61.4%) | 73 (64.0%) | 110 (59.8%) | 0.464 |
| Hyperuricemia | 50 (16.8%) | 20 (17.5%) | 30 (16.3%) | 0.780 |
| Chronic kidney disease | 59 (19.8%) | 18 (15.8%) | 41 (22.3%) | 0.172 |
| Atrial fibrillation | 17 (5.7%) | 5 (4.4%) | 12 (6.5%) | 0.440 |
| Current smoker | 203 (68.4%, n = 297) | 81 (71.7%, n = 113) | 122 (66.3%) | 0.333 |
| Haemodialysis on admission | 9 (3.0%) | 2 (1.8%) | 7 (3.8%) | 0.315 |
| History of previous myocardial infarction | 22 (7.4%) | 6 (5.3%) | 16 (8.7%) | 0.271 |
| History of chronic heart failure | 5 (1.7%) | 1 (0.9%) | 4 (2.2%) | 0.397 |
| History of previous PCI | 29 (9.7%) | 5 (4.4%) | 24 (13.0%) | 0.014 |
| Killip class | | | | |
| I or II | 253 (86.1%, n = 294) | 93 (82.3%, n = 113) | 160 (88.4%, n = 181) | 0.142 |
| III or IV | 41 (13.9%, n = 294) | 20 (17.7%, n = 113) | 21 (11.6%, n = 181) | |
| Shock on admission | 27 (9.1%) | 16 (14.0%) | 11 (6.0%) | 0.019 |
| Systolic blood Pressure on admission, (mmHg) | 132 ± 25 (n = 293) | 133 ± 27 (n = 111) | 130 ± 24 (n = 182) | 0.430 |
| Diastolic blood Pressure on admission, (mmHg) | 75 ± 18 (n = 289) | 79 ± 19 (n = 109) | 74 ± 17 (n = 180) | 0.014 |
| Heart rate on admission, (bpm) | 77 ± 19 | 78 ± 17 | 78 ± 20 | 0.998 |
| Peak creatine kinase, (U/L) | 1647 ± 2419 | 2570 ± 2813 | 1074 ± 1937 | <0.001 |
| Target vessel | | | | |
| RCA | 90 (30.2%) | 40 (35.1%) | 50 (27.2%) | 0.148 |
| LCA | 208 (69.8%) | 74 (64.9%) | 134 (72.8%) | |
| Left ventricular ejection fraction < 40% | 31 (10.8%, n = 288) | 17 (15.2%, n = 112) | 14(8.0%, n = 176) | 0.078 |
| Medication at discharge | | | | |
| Beta blocker | 223 (78.0%, n = 286) | 99 (90.8%, n = 109) | 124 (70.1%, n = 177) | <0.001 |
| ACE- inhibitor | 169 (59.1%, n = 286) | 82 (75.2%, n = 109) | 87 (49.2%, n = 177) | <0.001 |
| ARB | 76 (26.6%, n = 286) | 14 (12.8%, n = 109) | 62 (35.0%, n = 177) | <0.001 |
| MRA | 31 (10.8%, n = 286) | 15 (13.8%, n = 109) | 16 (9.0%, n = 177) | 0.242 |
| Statin | 260 (90.9%, n = 286) | 100 (91.7%, n = 109) | 160 (90.4%, n = 177) | 0.833 |

Data are expressed as mean ± SD or number (percentage). Pearson's chi-square test was used for categorical variables, normally distributed continuous variables were compared using Student's t-test, and the Mann-Whitney U test was used for abnormally distributed continuous variables.

PCI, percutaneous coronary intervention; RCA, right coronary artery; LCA, left coronary artery; ACE, angiotensin-converting enzyme; ARB, angiotensin II receptor blocker; MRA, mineralocorticoid receptor antagonist.

to 4.29, p = 0.032) after controlling for actual age, gender, peak creatine kinase, and history of heart failure (versus younger vascular age group).

## Discussion

In this study, we developed and validated a deep learning algorithm based on a 2D-CNN for the age prediction using CAG images. We demonstrated that the predicted age had promising potential for predicting patient outcomes, and we also showed which coronary artery feature

**Table 3. Comparison of clinical outcomes between the younger vascular age group and older vascular age group.**

| Variables | All (n = 298) | Younger vascular age group (n = 114) | Older vascular age group (n = 184) | P value |
|---|---|---|---|---|
| MACE (cardiac death, ACS, non-fatal cerebral infarction, admission for heart failure), n (%) | 55 (18.5%) | 14 (12.3%) | 41 (22.3%) | 0.031 |
| Cardiac death, n (%) | 17 (5.7%) | 5 (4.4%) | 12 (6.5%) | 0.432 |
| ACS, n (%) | 14 (4.7%) | 5 (4.4%) | 9 (4.9%) | 0.841 |
| Non-fatal cerebral infarction, n (%) | 16 (5.4%) | 2 (1.8%) | 14 (7.6%) | 0.029 |
| Admission for heart failure, n (%) | 17 (5.7%) | 4 (3.5%) | 13 (7.1%) | 0.198 |

Data are expressed as number (percentage). Pearson's chi-square test was used for categorical variables.

MACE, major adverse cardiac events; ACS, acute coronary syndrome

most influenced the predictions of the neural network in the Grad-CAM analysis. MACE were more frequently observed in the older vascular age group than in the younger vascular age group (p = 0.017). Furthermore, the older vascular age group was significantly associated with MACE (hazard ratio 2.14, 95% CI 1.07 to 4.29, p = 0.032) after controlling for known clinical risk factors. To the best of our knowledge, this is the first study to develop a neural network for predicting age based on CAG imaging and to use Grad-CAM to demonstrate the coronary artery feature that may be essential for the neural network decision-making.

Neural networks have been applied to data from various modalities for the purposes of age prediction [10, 26]. In the field of cardiology, neural networks for predicting chronological age from ECGs or chest x-rays have been developed and applied to clinical studies to investigate the clinical implications of predicted age. In such a study, a model for predicting chronological age from ECGs showed an R squared value of 0.70 and predicted age was found to be

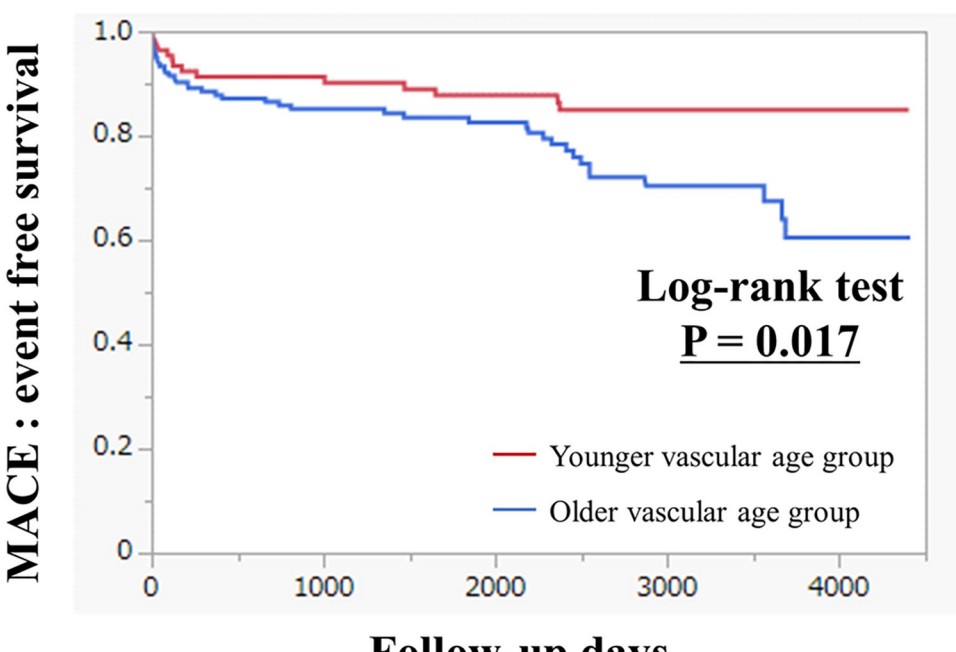

**Fig 5. Kaplan-Meier curves for major adverse cardiovascular events (MACE) in the younger vascular age group and older vascular age group.**

**Table 4. Multivariate Cox hazards analysis to predict MACE.**

| Independent variables | Dependent variable: MACE | | |
|---|---|---|---|
| | Hazard ratios | 95% confidence interval | P value |
| Older vascular age group (vs. younger vascular age group) | 2.14 | 1.07–4.29 | 0.032 |
| Actual age (per one year increase) | 1.04 | 1.01–1.07 | 0.007 |
| Male sex (vs. female sex) | 1.19 | 0.62–2.30 | 0.600 |
| Peak creatine kinase (per 1,000 U/L increase) | 1.23 | 1.11–1.35 | <0.001 |
| History of heart failure admission | 3.74 | 0.90–15.6 | 0.071 |

MACE, major adverse cardiac events.

associated with patient comorbidity [10]. Another study created a model for predicting age from chest x-rays and showed R squared of 0.92 and found that predicted age was associated with outcomes of patients with heart failure [26]. However, to the best of our knowledge, so far, there was no study that attempted to predict age from CAG images, which involves a set of images containing time series information. Our neural network to predict age from CAG images showed an R squared of 0.72, and we believe that its accuracy is sufficiently high compared with previous studies. Furthermore, while most previous studies used training data with more than 100,000 samples to create their models, our neural network was trained only using several thousand videos. This suggests that the method of treating the video on a frame-by-frame basis and then finally averaging the output of the neural network may have contributed to efficient extraction of information contained in the CAG video [20]. It also suggests that the age-related imaging features contained in the CAG imaging are robust.

Although several studies have proposed CAC scoring on coronary CT [27, 28] (estimating the degree of coronary artery calcification) as an indicator of aging, and vascular tortuosity is known to show age-related change in CAG [29], it had not been clearly established which CAG imaging features allowed age prediction. In addition, it is particularly important to identify clinical findings that are potentially modifiable. In this study, we used the Grad-CAM method to provide the visualization of the CAG image regions that may be essential for age prediction by the neural network and found that the neural network may focus on the limbus of the coronary arteries, rather than on local coronary artery features. This result may suggest that the neural network is correctly extracting information from the coronary artery images by themselves (depending on the neural network training, the neural network could extract information from the ribs, lung fields, and cardiac shadow). It is also possible that the coronary artery limbus may contain age-related information, which was not previously reported.

In our analysis of ACS patients who underwent PCI at our institution, we showed that when the patients were stratified according to the vascular age estimated from pre-PCI CAG images, the stratification showed a significant association with long-term outcomes. This suggests that pre-treated coronary artery status may provide useful information reflecting the patient prognosis. In general, the training of a neural network requires a large quantity of labelled data and the cost of labelling sufficient numbers of data entries to create the network can be enormous. However, the method used in this study to extract imaging features relevant to age, which has a well-established association with prognosis [30, 31], may have a potential for creating prognostic models.

The potential clinical implications of the present study should be noted. The age predicted using CAG imaging by neural network had high predictive value. Our study is particularly relevant because it adds further value to CAG and may lead to the exploration of new clinical findings that are potentially modifiable. For example, there may be a method for detecting

changes in vascular age over time to assist in determining drug therapy, although this could not be assessed in the present study. Future studies to assess the impact of age-related coronary artery features on the clinical practice are warranted.

The present study has the following limitations. First, because this was a single-centre retrospective study, there may have been a patient selection bias, which makes it difficult to generalize our results. Furthermore, as the model was validated with separate internal data, it is possible that the model performance might be lower with external data. Second, in the present study, there is a possibility that the neural network model might be trained on biased data because of the lack of information on comorbidities or the reason for CAG. Since we enrolled patients undergoing standby diagnostic CAG for any reason to train the model, there might be no controls, in other words, "healthy" subjects. We also validated the impact of predicted age on clinical outcomes of ACS patients, whereas the data used to train the neural network did not focus on ACS. Ideally, an age prediction by the neural network trained on coronary arteries of ACS patients should be developed, which seems to be difficult because of the limited number of ACS cases. In addition, the cut off age for younger and older vascular age groups was set at 65 years old, based on the average age of the ACS patients included in this study. It is difficult to generalize the method used in this study because, as mentioned above, it is the result of a single-centre retrospective study with a limited number of patients. In fact, the average age for PCI patients in Japanese clinical practice is known to be higher [32]. In the future, the usefulness of age predicted from CAG images should be explored using a large amount of CAG images and clinical data from ACS patients. Finally, although our Grad-CAM analysis showed that the neural network focused on the limbus of the coronary artery for both the LCA and RCA, we could not clarify exactly what feature of the coronary arteries is essential for predicting vascular age from CAG images. Such a problem is inherent to the nature of deep learning, which is often called the "black box of deep neural network" [33], and technologies that can explain deep neural network criteria in more detail are required.

## Conclusions

We developed a neural network to predict age based on CAG imaging and found that it showed high predictive value. The age predicted from CAG images by deep neural network could have significant associations with clinical outcomes in patients with ACS.

## Supporting information

**S1 Fig. Receiver operating characteristics (ROC) analysis of age classification.** Shown is the ROC curve for age classification ($\geqq$ 65 years old) in the test dataset. The overall area under the curve (AUC) was 0.839.
(TIF)

## Acknowledgments

We thank Edanz (https://jp.edanz.com/ac) for editing a draft of this manuscript.

## Author Contributions

**Conceptualization:** Shinnosuke Sawano.

**Data curation:** Shinnosuke Sawano, Masataka Sato, Susumu Katsushika, Hiroki Shinohara, Atsushi Kobayashi, Kazutoshi Hirose, Tatsuya Kamon, Hiroyuki Kiriyama.

**Formal analysis:** Shinnosuke Sawano.

**Investigation:** Shinnosuke Sawano, Hirotoshi Takeuchi.

**Methodology:** Shinnosuke Sawano.

**Project administration:** Shinnosuke Sawano.

**Supervision:** Satoshi Kodera, Issei Komuro.

**Validation:** Satoshi Kodera.

**Visualization:** Shinnosuke Sawano.

**Writing – original draft:** Shinnosuke Sawano.

**Writing – review & editing:** Satoshi Kodera, Masataka Sato, Susumu Katsushika, Issei Sukeda, Hirotoshi Takeuchi, Hiroki Shinohara, Atsushi Kobayashi, Hiroshi Takiguchi, Kazutoshi Hirose, Tatsuya Kamon, Akihito Saito, Hiroyuki Kiriyama, Mizuki Miura, Shun Minatsuki, Hironobu Kikuchi, Yasutomi Higashikuni, Norifumi Takeda, Katsuhito Fujiu, Jiro Ando, Hiroshi Akazawa, Hiroyuki Morita, Issei Komuro.

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
