## [Decision Letter · Decision Letter 0]

14 Sep 2022

PONE-D-22-20987Age prediction from coronary angiography using a deep neural network : age as a potential label to extract prognosis-related imaging features.PLOS ONE

Dear Dr. Kodera

Thank you for submitting your manuscript to PLOS ONE. After careful consideration, we feel that it has merit but does not fully meet PLOS ONE’s publication criteria as it currently stands. Therefore, we invite you to submit a revised version of the manuscript that addresses the points raised during the review process.

We look forward to receiving your revised manuscript.

Kind regards,

Xianwu Cheng, M.D., Ph.D., FAHA

Academic Editor

PLOS ONE

Journal Requirements:

Reviewers' comments:

Reviewer's Responses to Questions

**Comments to the Author**

1. Is the manuscript technically sound, and do the data support the conclusions?

Reviewer #1: Yes

Reviewer #2: Yes

2. Has the statistical analysis been performed appropriately and rigorously? 

Reviewer #1: Yes

Reviewer #2: Yes

3. Have the authors made all data underlying the findings in their manuscript fully available?

Reviewer #1: Yes

Reviewer #2: Yes

4. Is the manuscript presented in an intelligible fashion and written in standard English?

Reviewer #1: Yes

Reviewer #2: Yes

5. Review Comments to the Author

Reviewer #1: The authors evaluated whether a deep learning neural network could estimate vascular age based on coronary angiographic imaging. They also confirmed the clinical usefulness of the age prediction.

The findings are of clinically significance and have an impact.

However, several issues should be resolved.

Specific comments:

The authors enrolled patients undergoing standby diagnostic CAG for any reason.

In the study, there might be no control patients, in other words, healthy subjects. This may be a limitation of the study.

Methods: CAGs with PCI and CABG were excluded in the study. Does this mean that patients undergoing PCI or CABG after index CAG were excluded? Or were only patients with ad-hoc PCI excluded? Please clarify.

Methods, ‘Only those CAGs evaluated as “without clinically meaningful coronary stenosis (defined as 75% in at least one 9 branch)” were included. ‘: Even if coronary stenosis <75%, it may be the reason for cardiac ischemia. Therefore, FFR, iFR, scintigraphy, and so on are needed in the clinical settings.

In the study, 65 y/o was a cut off age for younger and older vascular age groups. However, in Japanese clinical practice, average age for PCI seems much older (Cardiovasc Interv Ther. 2022 Apr;37(2):243-247.). Please refer it and discuss.

Table 4: As a predictive value, LVEF is an important factor after onset of ACS. In addition, how did the authors adjust medical treatments to predict MACE?

Reviewer #2: I have no competing interests.

Interesting paper-

òprobably it appears a buit out of focus to use deep learning to derive vascular age

"gold standard" for vasciular age shoul,d be added

Impact of vascular age on outcomes should be added

Practical and clinical implications of vascular age should be added

6. PLOS authors have the option to publish the peer review history of their article (what does this mean?). If published, this will include your full peer review and any attached files.

Reviewer #1: **Yes: **Hideki ISHII

Reviewer #2: **Yes: **Fabrizio D'Ascenzo

---

## [Author Response · Author response to Decision Letter 0]

21 Sep 2022

Response to the comment from the Reviewer #1.

Thank you for your careful review of our manuscript. We have attempted to incorporate your valuable suggestion. We believe that your suggestion has significantly improved the overall scientific content of this work. Response to the specific question is given below:

Specific comment1:

The authors enrolled patients undergoing standby diagnostic CAG for any reason.

In the study, there might be no control patients, in other words, healthy subjects. This may be a limitation of the study.

Response 1. We agree with reviewer 1. The training data for our model may not have included healthy subjects. We revised the following sentence in the study limitations according to the reviewer #1’s suggestion:

“Since we enrolled patients undergoing standby diagnostic CAG to train the model for any reason, there might be no control patients, in other words, healthy subjects.” (Please see line 20 of page 17 and line1 of page 18.)

Specific comment2:

Methods: CAGs with PCI and CABG were excluded in the study. Does this mean that patients undergoing PCI or CABG after index CAG were excluded? Or were only patients with ad-hoc PCI excluded? Please clarify.

Response 2.

We agree with reviewer 1. Patients undergoing PCI or CABG after index CAG were included in the study. We excluded patients who underwent ad-hoc PCI. If for some reason a wire or balloon used for PCI or measurement of fractional flow reserve was captured on the images, these images were manually excluded, as were contrast images of coronary artery bypass. We revised the following sentence in the study Methods according to the reviewer #1’s suggestion:

“Consecutive patients aged ≥18 years who underwent standby diagnostic CAG for any reason between January 2010 and December 2015 at The University of Tokyo Hospital were reviewed. Patients undergoing PCI or CABG after index CAG were included in the study. Patients who underwent PCI concurrently diagnostic catheterization (i.e., ad-hoc PCI) were excluded. CAGs that a highly trained cardiologist judged as showing insufficient contrast effect were excluded after image processing, and at the same time, if for some reason a wire or balloon used for PCI or measurement of fractional flow reserve was captured on the images, these images were manually excluded, as were contrast images of coronary artery bypass.” (Please see line 1-8 of page 4.)

Specific comment3:

Methods, ‘Only those CAGs evaluated as “without clinically meaningful coronary stenosis (defined as 75% in at least one branch)” were included. ‘: Even if coronary stenosis <75%, it may be the reason for cardiac ischemia. Therefore, FFR, iFR, scintigraphy, and so on are needed in the clinical settings.

Response 3.

We agree with reviewer 1. FFR, iFR, scintigraphy, and so on are needed in the clinical settings to consider clinically meaningful coronary stenosis. In this study, it was difficult to obtain FFR, iFR, and scintigraphy information, so we revised following sentence in the method in order to accurately present the reality of this study.

“Only those CAGs evaluated as “without coronary artery stenosis greater than 75%” were included.” (Please see lines 9-10 of page 4.)

Specific comment4:

In the study, 65 y/o was a cut off age for younger and older vascular age groups. However, in Japanese clinical practice, average age for PCI seems much older (Cardiovasc Interv Ther. 2022 Apr;37(2):243-247.). Please refer it and discuss.

Response 4.

We agree with reviewer 1. The average age for PCI is higher in Japanese clinical practice. In this single-center study, the mean age of patients was 67 years, and the cutoff was set at 65 years, which is younger than in Japanese epidemiological studies (32, Cardiovasc Interv Ther. 2022 Apr;37(2):243-247.). The results of this study are difficult to generalize, as noted in the limitations. In the future, we would like to examine the cutoff and usefulness of the method proposed in this study more deeply using National Registry data. We revised the following sentence in the study Methods according to the reviewer #1’s suggestion. We believe that your suggestion has significantly improved the discussion of this work.

“In addition, the cut off age for younger and older vascular age groups was set at 65 years old, based on the average age of the ACS patients included in this study. It is difficult to generalize the method used in this study because, as already mentioned, it is the result of a single-centre retrospective study with a limited number of patients. In fact, the average age for PCI patients in Japanese clinical practice is known to be higher. In the future, it may be useful to examine the usefulness of predictive age from coronary angiography by large amount of CAG images and clinical data from ACS patients.” (Please see lines 4-10 of page 18.)

Specific comment5:

Table 4: As a predictive value, LVEF is an important factor after onset of ACS. In addition, how did the authors adjust medical treatments to predict MACE?

Response 5.

We agree with reviewer 1. Information on LVEF and medications at discharge for ACS patients was added to table 2; unfortunately, multivariate analysis including LVEF and medications did not show a significant association between predicted age and prognosis. The sample size of this study was very small, which may have limited its power, and further studies using larger data are warranted. The example of the results of the multivariate analysis will be submitted as " Response to Reviewers.docx".

(Please see Response table1 and Response table2 in " Response to Reviewers.docx".)

Response to the comment from the Reviewer #2.

Thank you for your careful review of our manuscript. We have attempted to incorporate your valuable suggestion. We believe that your suggestion has significantly improved the overall scientific content of this work. Response to the specific question is given below:

Specific comment1:

òprobably it appears a buit out of focus to use deep learning to derive vascular age

"gold standard" for vasciular age shoul,d be added

Response 1.

Thank you for your careful comment. The gold standard of the vascular age is evaluated as coronary artery calcium (CAC) scanning by computed tomography (CT) and carotid intima-media thickness (CIMT) assessment by B-mode ultrasonography (Cuocolo A, Klain M, Petretta M. Coronary vascular age comes of age. J Nucl Cardiol. 2017 Dec;24(6):1835-1836. doi: 10.1007/s12350-017-1078-6. Epub 2017 Oct 3. PMID: 28975506.). In our study, we used deep learning to analyze from a different perspective (age-related changes in coronary angiography) from the gold standard of vascular age. To show the gold standard for vascular age we describe in introduction as following sentence.

“Vascular age is a concept in relation to the hypothesis that the conversion of chronological age to age derived from vascular imaging features will lead to more accurate assessment of an individual’s cardiovascular risk. Although coronary artery calcium (CAC) scoring by computed tomography (CT)(6) and carotid intima-media thickness (CIMT)(7) assessment by B-mode ultrasonography can be used to define vascular age(8), it is not clearly known whether CAG contains useful age-related imaging features.” (Please see lines 7-12 of page 3.)

Specific comment2:

Impact of vascular age on outcomes should be added

Response 2.

Thank you very much for your comment. In this study, the clinical prognosis may be worse in the older vascular age group predicted from CAG images. Patients undergoing coronary angiography are expected to be at high risk of arteriosclerosis, but age-related information obtained from coronary arteries may be used to assess these patient risk. To show the impact of vascular age on outcome we describe in discussion as following sentence.

“In our analysis of ACS patients who underwent PCI at our institution, we showed that when the patients were stratified according to the vascular age estimated from pre-PCI CAG, the stratification showed a significant association with long-term outcomes. This suggests that pre-treated coronary artery status may provide useful information reflecting the patient prognosis. In general, the training of a neural network requires a large quantity of labelled data and the cost of labelling sufficient numbers of data entries to create the network can be enormous. However, the method used in this study to extract prognosis-related features from images on the basis of age, which has a well-established association with prognosis (30, 31), may have a potential for creating prognostic models.” (Please see lines 1-8 of page 17.)

Specific comment3:

Practical and clinical implications of vascular age should be added

Response 3.

Thanks very much for your comment. Since this study is a single-center retrospective study, it is difficult to generalize the findings obtained in this study immediately, but we consider the following opinions as a proof of concept.

“The potential clinical implications of the present study should be noted. The age predicted using CAG imaging by neural network had high predictive value. Our study is particularly relevant because it adds further value to CAG and may lead to the exploration of new clinical findings that are potentially modifiable. For example, there is a method such as observing changes in vascular age over time to assist in determining drug therapy and follow-up periods, although this could not be assessed in the present study. Future studies to assess the impact of age-related coronary artery features in the clinical practice are warranted.” (Please see lines 9-14 of page 17.)

---

## [Decision Letter · Decision Letter 1]

17 Oct 2022

Age prediction from coronary angiography using a deep neural network : age as a potential label to extract prognosis-related imaging features.

PONE-D-22-20987R1

Dear Dr. Kodera

We’re pleased to inform you that your manuscript has been judged scientifically suitable for publication and will be formally accepted for publication once it meets all outstanding technical requirements.

Kind regards,

Xianwu Cheng, M.D., Ph.D., FAHA

Academic Editor

PLOS ONE

Additional Editor Comments (optional):

All original concerns have been addressed by the authors.

Reviewers' comments:

Reviewer's Responses to Questions

**Comments to the Author**

1. If the authors have adequately addressed your comments raised in a previous round of review and you feel that this manuscript is now acceptable for publication, you may indicate that here to bypass the “Comments to the Author” section, enter your conflict of interest statement in the “Confidential to Editor” section, and submit your "Accept" recommendation.

Reviewer #1: All comments have been addressed

Reviewer #2: All comments have been addressed

2. Is the manuscript technically sound, and do the data support the conclusions?

Reviewer #1: Yes

Reviewer #2: (No Response)

3. Has the statistical analysis been performed appropriately and rigorously? 

Reviewer #1: Yes

Reviewer #2: (No Response)

4. Have the authors made all data underlying the findings in their manuscript fully available?

Reviewer #1: Yes

Reviewer #2: (No Response)

5. Is the manuscript presented in an intelligible fashion and written in standard English?

Reviewer #1: Yes

Reviewer #2: (No Response)

6. Review Comments to the Author

Reviewer #1: Thank you for changings.

The reviewer found that the manuscript has been significantly improved.

There is no criticism.

Reviewer #2: (No Response)

7. PLOS authors have the option to publish the peer review history of their article (what does this mean?). If published, this will include your full peer review and any attached files.

Reviewer #1: **Yes: **Hideki ISHII

Reviewer #2: **Yes: **Fabrizio D'Ascenzo

---

## [Editor Report · Acceptance letter]

20 Oct 2022

PONE-D-22-20987R1 

Age prediction from coronary angiography using a deep neural network: age as a potential label to extract prognosis-related imaging features. 

Dear Dr. Kodera:

I'm pleased to inform you that your manuscript has been deemed suitable for publication in PLOS ONE. Congratulations! Your manuscript is now with our production department. 

Kind regards, 

on behalf of

Associate Prof. Xianwu Cheng 

Academic Editor

PLOS ONE